# Effects of Crop Resistance on the Tritrophic Interactions between Wheat Lines, *Schizaphis graminum* (Hemitera: Aphididae), and *Propylaea japonica* (Coleoptera: Coccinellidae)

**DOI:** 10.3390/plants11202754

**Published:** 2022-10-18

**Authors:** Xiang-Shun Hu, Jing-Wen Li, Jing-Feng Peng, Han Wang, Fan-Ye Yan, Zi-Fang Zhou, Zhan-Feng Zhang, Hui-Yan Zhao, Yi Feng, Tong-Xian Liu

**Affiliations:** 1State Key Laboratory for Crop Stress Biology in Arid Areas and Key Laboratory of Crop Pest Management on the Northwest Loess Plateau, Ministry of Agriculture, College of Plant Protection, Northwest A&F University, Xianyang 712100, China; 2College of Agriculture, Guizhou University, Guiyang 550025, China

**Keywords:** *Propylaea japonica*, *Schizaphis graminum*, life-history traits, functional response, prey consumed, proportion of prey consumed, predatory efficiency

## Abstract

Crop resistance and biological control are both considered efficient and environmentally friendly methods of sustainable pest control. In this study, we aimed at investigating the direct influence of four wheat lines with varying resistance level on the life-history traits of the greenbug, *Schizaphis graminum*, and the mediational effect on the functional response of a predatory ladybird, *Propylaea japonica*, under laboratory conditions. Results showed that the aphid fitness was the lowest for aphids that had been feeding on wheat line ‘98-10-19’ for one year. These aphids had the longest development time, and least adult mass, minimal mean relative growth rate, and lowest reproductive fitness. In contrast, the aphids that fed on wheat line ‘98-10-30’ were the fittest, with the shortest development time and highest levels of reproductive fitness. The predatory activities of the ladybeetle, especially the adult male significantly decreased following the consumption of aphids belonging to the ‘98-10-19’-acclimated population. However, there were no significant differences in predatory efficiency (net attack frequency) among the four aphid acclimated populations. Our results showed that the wheat line ‘98-10-19’ has a relative higher resistance to *S. graminum* than the other three wheat lines, which could further decrease the amount of prey available for consumption. However, the ecological effect of the resistance of ‘98-10-19’ to *S. graminum* posed no negative influence on the biocontrol potential of *P. japonica* to these aphids, as their predatory efficiency increases at the fourth instar larvae phase.

## 1. Introduction

Tripartite interactions among crop host, insect pest, and its natural enemies are complicated [1,2]. Crop resistance and biological control are both considered as efficient and environmentally friendly ways to control pest sustainably [3,4,5]. Crop resistance is a basic measure of integrated pest management (IPM) and ecological pest management (EPM) [1,2,6]. In the history of crop pest control, crop resistance has been widely applied and is used in the management of a series of pests, such as the Hessian fly (*Mayetiola destructor*), wheat midge (*Sitodiplosis mosellana* and *Comtarinia tritci*), rice brown planthopper (*Nilaparvata lugens*), cotton bollworm (*Helicoverpa armigera*), grape phylloxera (*Viteus vitifoliae*), and apple cotton aphid (*Eriosoma lanigerum*) [1]. Biological control agents play an important role in ecological security and the control of insect pest in agroecosystems and are, thus, considered as an effective technique that uses nonchemical and environmentally friendly methods [7]. The interactions between crop resistance and biological control are presently an area of growing research interest in integrated pest management (IPM) strategy [8,9,10]. However, combining natural enemies and plant resistance may enhance or reduce the synergistic effect to control insect pests. The resistance of host crop merely reduces the pest suitability, with largely minor effects on prey suitability for predators in an agroecosystem [11,12]. Some empirical evidence has shown that crop resistance and natural enemy jointly synergize to affect the herbivore population growth. Furthermore, other evidence has emerged regarding many successful cases of natural biological control having been linked to the development and release of resistant varieties. For example, a synergistic effect was found between the host plant resistance in wheat and predators *Hippodamia variegata* (Goeze) in the integrated pest management of the Russian wheat aphid, *Diuraphis noxia* (Kurdjumov) [12]. However, some reports have shown that crop varieties (lines) indirectly affect predators through their prey. Kersch-Becker & Thaler found that the plant resistance of high-resistance tomato, *Solanum lycopersicum* L., reduced the predation rate (consumptive effect) of *H**. convergens* to aphids *Macrosiphum euphorbiae* [10]. They further showed that the natural predator’s colonization was diverse and abundant, and their relative consumption rates were higher for aphid on low-resistance tomato than those that fed on high-resistance tomato [13]. Shannag and Obeidat reported that the seven-spot ladybird beetle, *Coccinella septempunctata* L., feeding on the aphids, *Aphis fabae*, from the partially resistant cultivar had a prolonged embryonic larval developmental time, which is the duration required from egg laying to adult emergence, and a decreased female fecundity and fertility compared with that of feeding the aphids from susceptible cultivar. However, these negative effects of resistance through prey are not always effective for the reduction in aphid numbers [11]. Hence, a better understanding of the tripartite interaction and perceptive insight into the workings of nature are necessary to predict how to combine natural enemies and plant resistance for sustainable pest control [14,15].

Wheat (*Triticum aestivum* L.) is a staple food for more than one third of the global population and is cultivated worldwide [16]. Three cereal aphid species, the greenbug, *Schizaphis graminum* Rondani, the grain aphid *Sitobion miscanthi* Takahashi, (widely misreported as *S. avenae* (Fab.) in China), and the bird cherry-oat aphid *Rhopalosiphum padi* L., are common pests that attack wheat plants in China. The greenbug *S. graminum* is a destructive pest of wheat and 70 other graminaceous species planted in arid and semi-arid areas worldwide. This aphid probes and sucks the phloem sap, and invariably poses as a vector for transmitting plant viruses, e.g., barely yellow dwarf virus [9,17,18,19]. Chemical pesticides have been used to control the cereal aphids and, consequently, pose serious ecological and environmental pollution risk. Resistant wheat varieties (lines) have proven to be the most economical, efficient, and environmentally friendly approach to control cereal aphids because they are less expensive and do not have destructive effects on the natural enemies of these pests and agroecosystems [20,21]. The ladybeetle, *Propylaea japonica* (Thunberg) (Coleoptera: Coccinellidae), is considered a successful natural enemy because of its tolerance to high temperatures and insecticides in arid and semi-arid areas in East Asia [2,22,23]. Basically, it preys on some small insect pests, such as aphids, whiteflies, and mates in agricultural systems, and is used as an indigenous biological control agent essential in the innovation and development of integrated pest management in China [22,24]. 

Cereal aphids are the main prey of ladybeetles in wheat fields. These aphids are parthenogenetic in the growing season of their host plants and have telescoping generations where the granddaughters of a female aphid are already developing within the daughters’ body [25]. The maternal effect (maternal environmental effects on offspring) on the cereal aphids is strongly influenced by the resistance of maternal host wheat varieties (lines) over time [26,27,28]. The resistant wheat variety (line) could effectively slow the rates of aphid development, reproduction, and population growth, as well as further affect the predator via prey quality [9]. However, how host plant resistance regulates the foraging efficiency of natural enemies via herbivores over time is not well understood. In the present study, four wheat lines (two pairs of synthetic sister wheat lines) were chosen to explore the feasibility of combining wheat line resistance characteristics with a predator to control greenbug. Notably, these wheat lines have different resistance to *Sitobion avenae* (Fab.) [29]. However, the resistance statuses of these four wheat lines to greenbugs have not been investigated. In this study, we compared the life traits of greenbug populations that acclimated to these wheat lines, and the functional responses of *P. japonica* that prey on these aphid acclimated populations. We hope that the information from crop–aphid–natural predatory interactions could help us better understand the tripartite interaction in agroecosystems, as well as the community evolution processes in wheat fields that have shaped them. 

## 2. Results

### 2.1. The Life-History traits of Four Wheat-Acclimated Aphid Populations

The life-history traits of four wheat-acclimated greenbug populations that acclimated to the four wheat lines for more than a whole year are shown in Table 1.

The weight of the newborn nymph aphids, W1st, was not significantly different among the four wheat-acclimated aphid populations (*F* = 1.97, *df* = 3, 116. *p* = 0.12). However, the adult aphid weight, Wa, was significantly different among the four wheat-acclimated aphid populations (*F* = 9.09, *df* = 3, 916. *p* < 0.001). The Wa of the ‘98-10-30’-acclimated aphid population was significantly heavier (22.67–51.74%), and the Wa of the ‘98-10-19’-acclimated aphid population was significantly lower (19.08–34.10%) than that of the other three wheat-acclimated aphid populations (*p* < 0.001), and the adult aphid weight Wa of the ‘186Tm39’ and ‘186Tm47’-acclimated aphid populations was not significantly different. The development time of the ‘98-10-30’-acclimated aphid population was significantly longer (more than 0.5 d) than that of the other three wheat-acclimated aphid populations (*p* < 0.001), and the mean relative growth rate (MRGR), intrinsic rates of natural increase (r_m_), and net reproduction rate (NRR) of the ‘98-10-30’-acclimated aphid population were significantly less (18.33–23.31%, 21.32–24.02%, and 25.65–42.62%, respectively) than those of the other three populations (*p* < 0.001), and these traits were not significantly different among the other three wheat-acclimated aphid populations. The fecundity of the ‘98-10-30’-acclimated aphid population was greater than that of the ‘98-10-19’ (65.22%) and ‘186Tm39’ (38.97%) acclimated aphid populations, and the fecundity of the ‘186Tm47’-acclimated aphid population was also greater than that of the ‘98-10-19’-acclimated aphid population (43.92%). The nymph survival rates (NSR) ranged between 0.70 and 0.90 but the differences among wheat-acclimated aphid populations were not significant (*F* = 2.11, *df* = 3, 116. *p* = 0.102). 

Age-stage survival rates are shown in Figure 1. This gives the probability that a newborn will survive to the next day (same line) and the next stage (different line). The rapid decrease in survival rate of nymph is shown in the curves of ‘98-10-19’ and ‘186Tm47’-acclimated aphid populations. In addition, a slow decrease in survival rate of nymph is shown in the curves of ‘98-10-30 and ‘186Tm39’-acclimated aphid populations. Most nymphs of the ‘98-10-19’-acclimated aphid population molt to adult on the ninth day; however, the other three acclimated aphid populations molt to adult on the eighth day.

These results showed that the fitness of greenbugs was significantly different among the four wheat lines over a year. This indicates that the four wheat lines have diverse resistance traits to *S. graminium*. The resistance of wheat line ‘98-10-19’ was the highest among the four wheat lines compared to the other three wheat lines. The weight gain and fecundity of the ‘98-10-30’-acclimated aphid population were significantly greater than those of the ‘186Tm47’ and ‘186Tm39’-acclimated aphid populations, indicating that ‘98-10-30’ was the most susceptible wheat line to greenbug among the four wheat lines.

### 2.2. The Prey Consumption and Proportion of Prey Consumed of Ladybird Feeding with Four Prey Wheat-Acclimated Aphid Populations

The wheat-acclimated aphid populations (wheat lines), prey densities, ladybird age stage, and the interaction of pairwise combinations were all significantly affected by prey consumption *N_e_*, as well as proportion of prey consumed *N_e_/N_0_* of the predatory, but the *N_e_/N_0_* interaction of all three combinations was not significant (Table 2).

The resistance of wheat lines affected the *Ne* and *N_e_/N_0_* of ladybeetles over time. Within a developmental stage, almost all of the differences in the prey consumption (*N_e_*) at different prey acclimated populations and various prey densities of *P. vulgaris* were significant (all of the *p*-values less than 0.001). 

When the data for all developmental stages were pooled together, the *N_e_* of *P. vulgaris* feeding on the aphids from the ‘98-10-19’-acclimated population (12.76 ± 1.05) was significantly less than that of the ‘98-10-30’-acclimated population (13.92 ± 1.15). The *Ne* from the ‘186Tm39’ (13.33 ± 1.07) and ‘186Tm47’ (13.34 ± 1.10) acclimated aphid populations in the middle were not significantly different from both the ‘98-10-19’- and ‘98-10-30’-acclimated aphid populations. The enemy *N_e_/N_0_* feeding on aphids from ‘98-10-19’-acclimated population (74.00 ± 2.37%) was significantly less than that from ‘98-10-30’ (77.55 ± 2.20%) and ‘186Tm47’ (77.38 ± 2.19%) acclimated populations; the *N_e_/N_0_* from the ‘186Tm39’ (75.61 ± 2.31%) acclimated population in the middle was not significantly different from the other three wheat-acclimated aphid populations.

When the data of four populations were pooled together, the *N_e_* and *N_e_/N_0_* of the fourth instar larvae (*N_e_* = 19.54 ± 1.37 and *N_e_/N_0_*
*=* 79.68 ± 1.54%) were greatest, and both female (*N_e_* = 21.27 ± 1.70 and *N_e_/N_0_*
*=* 78.78 ± 1.24%) and male adults (*N_e_* = 18.62 ± 1.38 and *N_e_/N_0_*
*=* 74.42 ± 1.58%) were greater, and all of these three age stages were significantly greater than that of the second larvae (*N_e_* = 4.79 ± 0.27 and *N_e_/N_0_*
*=* 55.22 ± 2.45%) and first instar larvae (*N_e_* = 1.98 ± 0.11 and *N_e_/N_0_*
*=* 49.79 ± 2.76%). The *N_e_* and *N_e_/N_0_* of third instar larvae (*N_e_* = 13.82 ± 0.83 and *N_e_/N_0_*
*=* 64.19 ± 1.97%) were in the middle. The *N_e_* and *N_e_/N_0_* of the different populations were the same as those pooled together (details are shown in Appendix A).

### 2.3. The Functional Responses of Ladybeetle Feeding with Four Prey Acclimated Populations

Significant negative linear terms were derived from logistic regressions for each developmental stages of predators feeding on all of the four wheat-acclimated aphid populations (details are shown in Appendix A). This indicates that all developmental stages of the predators displayed a type 2 functional response fitted with the random predator equation when they preyed on aphids from each of the four wheat-acclimated aphid populations.

Most functional response curves overlapped between prey aphids from different acclimated populations at each developmental stage predators (details are provided in Appendix A).

### 2.4. The Attack Rate, Handling Times, and Predatory Efficiency of Ladybird Feeding with Four Prey Acclimated Populations

The attack rate (*a*) of the predator was significantly different among the different developmental stages (*F* = 7.59, *df* = 5, 18, *p* < 0.001) but not feeding among the different wheat-acclimated aphid populations (*F* = 0.96, *df* = 3, 20, *p* = 0.44). The highest *a* was observed in the fourth instar larvae (0.38 ± 0.01). It is significantly higher than that of other age stages (all less than 0.16).

The handling times (*T_h_*) of the predator were significantly different among the different developmental stages (*F* = 23.67, *df* = 5, 18, *p* < 0.001) but not among the different wheat-acclimated aphid populations (*F* = 0.59, *df* = 3, 20, *p* = 0.63). The shorter *T_h_* included the female (0.34 ± 0.00 m), male (0.46 ± 0.00m), fourth instar larvae (0.53 ± 0.00 m), and third instar larvae (0.84 ± 0.01). The longest *T_h_* was the first instar larvae, and the *T_h_* of the second instar larvae was in the middle.

The predatory efficiency (or net attack frequency) *a/Th* of the predator was significantly different among different developmental stages (*F* = 21.38, *df* = 5, 18, *p* < 0.001) but not among different acclimated populations (*F* = 0.14, *df* = 3, 20, *p* = 0.93). The highest *a/Th* was in the fourth instar larvae, the higher *a/Th* was in both female and male adults, and the lowest *a/Th* was in the first, second, and third instar larvae. However, it was interesting that the highest predatory efficiency was that of the fourth instar larvae predator feeding on the ‘98-10-19’ population (Figure 2). The lowest *a/Th* of male ladybeetles was also observed in the ‘98-10-19’ population. This caused *a/Th* of the predator feeding on the ‘98-10-19’ population to not be different from that feeding on the other three populations.

The 95% CI of the attack rate *a* overlapped between each pair of acclimated populations at each developmental stage of the ladybird. Most 95% CIs of the handling time *Th* overlapped, besides the first instar larvae’s handling time from the ‘98-10-30’ population being separated and longer than that from the ‘186Tm39’ and ‘98-10-19’ populations, and the female adults’ handling time from the ‘98-10-30’ population being separated and longer than those from ‘186Tm47’ and ‘98-10-19’ populations (details are shown in Appendix A). 

## 3. Discussion

In this study, we found that the different wheat lines had diverse resistance traits to *S. graminium*. Resistance has a negative effect on the life-history traits of *S. graminium*, grain weight, development time, MRGR, fecundity, r_m_, and NRR. This could further decrease the prey and, invariably, the proportion of prey consumed by *P. japonica* to the resistant wheat-line-acclimated aphid population over a year. We also found that host resistance effects differed among the development stages and sexes of predators. However, this resistance does not hinder the biological control potential of *P. japonica* against *S. graminum*.

The resistance of wheat varieties (lines) to cereal aphid of different species is complex. Our previous research results have shown that the resistances of different wheat varieties (lines) have a stronger ‘trade-off’ for the English grain aphid, *Sitobion avenae*, compared to *S. graminum* [30]. The synthetic sister lines ‘98-10-19’ and ‘98-10-30’ were the cross progeny of wheat *T. aestivum* (var. Chris) and *T. turgidum*. The other sister lines, ‘186Tm39’ and ‘186Tm47’, were the progeny of the hybrid of wheat *T. aestivum* variety 186 and *T. monococcum*. Accordingly, ‘98-10-30’ is a wheat line that is relatively resistant to ‘98-10-19’, and ‘186Tm47’ is more relatively resistant than ‘186Tm39’ to the English grain aphid [29]. Interestingly, the resistance of ‘98-10-19’ to *S. graminum* was greater than that of ‘98-10-30’, even over the entire year. 

The plant–insect–natural enemy tri-trophic relationship is more likely to affect the population dynamics of *P. japonica* [24]. In this study, the *N_e_* and *N_e_/N_0_* of *P. japonica* in the whole age stage preying *S. graminium* population acclimated on ‘98-10-19’, which is the highest resistant wheat line, were the lowest; however, their biological control potential was not significantly different from that of the acclimated aphid populations on other wheat lines, indicating that the resistance of ‘98-10-19’ to *S. graminum* did not negatively affect the biological control potential of *P. japonica* to *S. graminum*.

The plant species could mediate the quality of prey insect that feeds on pest which inhabit on it. Based on nutrition, the prey quality is a key factor that affects the growth, development, and reproduction of predatory insects [31,32,33]. The nutritional factors (different fatty acids and subsequent calories nutritional levels) of *Medicago sativa* L. (cv. ‘OKO8’) and *Vicia faba* L. (cv. ‘Windsor’) could affect the fitness of *C. septempunctata* L. via the herbivore prey *Acyrthosiphon pisum* Harris (Homoptera: Aphididae) [34]. The volatile cues emitted from psyllids- and Las bacteria-infected citrus plants exhibit host searching behavior and efficiency of *P. japonica* [35]. In our study, the weight of the first instar aphid was not significantly different among the four acclimated populations; however, the adult weight gain and fecundity of the aphids belonging to the ‘98-10-19’ population were significantly lower than those of the other three populations. The predation amount (*N_e_*) and predation rate (*N_e_/N_0_*) of *P. japonica* (whole developmental stage pooled together) to the aphid belonging to ‘98-10-19’ population did not increase, inversely decreased, and the male adult ladybird had the main contribution for the *N_e_* and *N_e_/N_0_* decrease on ‘98-10-19’. This indicates that the decrease in food quality (aphid weight) did not compensate for the increase in quantity (*N_e_*). We assumed that the resistance factors of the wheat line ‘98-10-19’ to aphids and their natural enemies may not be nutritional restriction but toxic factors from secondary metabolites. 

The type II functional response has always been used to fit aphids and other insects consumed by *P. japonica* based on logistic regressions [36,37,38,39,40]. Our study is consistent with this still. Within every single developmental stage of ladybird prey on each of the four aphid-acclimated populations, the logistic regression had linear parameters (<0). When we checked the scatter diagram of the numbers of eaten aphids vs. aphids densities, we found that almost all of the asymptotic 95% CIs of the a and *T_h_* estimates did not include zero (not significantly different from zero), indicating that these data well match the pattern of type II functional response [41]. 

In this study, the *N_e_* and *N_e_/N_0_* of the fourth instar larvae, adult females, and males of *P. japonica* were the highest and did not differ significantly (Table 2). We found that the fourth instar larvae of *P. japonica* had the highest potential stage (highest *a/T_h_*) to control aphids (Figure 2). These results were consistent with those reported in the fourth instar larvae of *C. septempunctata* [42] and *Propylea dissecta* [43], and, like the female of *C. septempunctata* [44] and *Cydonia vicina* nilotica Muls [45], have more potential to control aphids. 

## 4. Materials and Methods

### 4.1. Wheat, Aphid, and Ladybird

Four wheat lines were used in this study. Two synthetic sister lines, ‘186Tm39’ and ‘186Tm47’, were the progeny of a hybrid of wheat *T. aestivum* variety 186 and *T. monococcum*. Two other synthetic sister lines, ‘98-10-19’ and ‘98-10-30’, were cross progenies of wheat *T. aestivum* (var. Chris) and *T. turgidum*. 

A single individual apterous of *S. graminum* and 10 adults of *P. japonica* (male and female evenly distributed) were collected from an experimental field of Northwest A&F University, Yangling, Shaanxi, China (34°17ʹ35ʺN, 108°4ʹ18ʺE). The aphids were cultured on wheat seedlings (var. Xinong 1376) to establish the stock population. The ladybirds were reared with the grain aphid, *Sitobion*
*miscanthi*, which fed on wheat seedlings (cv. Xiannong 1376). 

The rearing and subsequent experiment conditions were as follows: a 3 × 5 m plant growth chamber at 25 ± 0.5 °C (day) and 22 ± 0.5 °C (night), a photoperiod of L16 h: D8 h, 50 ± 10% relative humidity, and 11,000 LX light intensity. At the two-leaf stage (Feekes 1.2, 13 days after sowing), fresh wheat seedlings were used for aphid rearing.

### 4.2. Aphid Population Acclimated on 4 Wheat Lines

The aphids were transferred from the stock population to fresh wheat seedlings of the four wheat lines at the two-leaf stage (Feekes 1.2, 13 d after sowing) using a small brush to establish four independent acclimated populations in four separate cages (60 × 60 × 60 cm), which were covered with 200-mesh gauze. The wheat seedlings used for aphid population acclimation were planted in plastic pots (9 cm bottom diameter, 14 cm top diameter, 10 cm height) filled with a mixed potting medium consisting of sand, humus, and black loam at a ratio of 1:3:3. The density was ~40 plants per pot and was changed once every half a month. Fresh wheat seedlings at the two-leaf stage (13 d after sowing) were moved to the cage, and ~50 nymphs (first and second instars) were transferred from the old wheat seedlings to the newly introduced wheat seedlings. Thereafter, the old wheat seedlings were then moved out of the cage. Twelve months later, the aphids from each wheat-acclimated population were used for the subsequent experiments. 

### 4.3. The Life-History Traits of Four Wheat-Acclimated Aphid Populations

A one-factor factorial design experiment was conducted. The factor was the wheat-acclimated aphid population (among ‘98-10-19’-acclimated, ‘98-10-30’-acclimated, ‘186Tm39’-acclimated, and ‘186Tm47’-acclimated). An apterous adult aphid from acclimated population was transferred from stock seedlings to a test seedling (same wheat line) to produce nymphs for 12 h. Only one newborn nymph remained on a test seedling planted in a plastic pot (8 cm bottom diameter, 12 cm top diameter, 9 cm height). The mother aphids and other newborn nymphs were removed. A total of 30 young nymphs were prepared from 30 single test seedlings for each acclimated population. All the aphids and test seedlings were placed in a growth chamber. The life-history traits of individual immature nymph development, mortality, development time (from newborn nymph to adult), and weights of the newborn nymphs, as well as the newly molted adult, and number of all newborn nymphs of each adult within a duration that was the same as nymphs that developed to adulthood were monitored and recorded daily.

The age-stage survival rate (SR) was calculated and fitted based on the age-stage-structure matrix [46]. 

The nymph survival rate NSR = $\sum_{j=1}^{m}SR_{j}$ (*j* is age stage) and the development time (DT) = the time from birth to adult emergence. The weight gain (WG) = adult weight (Wa) − 1st instar larvae weight (W1^st^). The mean relative growth rate (MRGR) = (lnWa − lnW1^st^)/DT). The fecundity = offspring produced per female within a duration that was the same as DT after maturation. The intrinsic natural increase rates (r_m_) = 0.738 ln(fecundity)/(2 × DT). The net reproduction rate (NRR) = NSR × fecundity/(2 × DT)) [30,47,48].

### 4.4. Functional Response of Ladybeetle Preying on Aphid

The ladybird larvae hatching or molting within 12 h, as well as the adult emergence 7 days later, were used in the experiment, and only the fourth instar aphids from each wheat-acclimated population were used to avoid reproduction during the experiment. 

A three-factor factorial design experiment was performed to study the functional responses of ladybird prey to different acclimated aphids. The first factor was the acclimated populations (or the wheat lines). The second factor was the age stage of *P. japonica* (first, second, third, and fourth instar nymphs, and male and female adults). The third factor was prey (aphid) densities, *N_0_*, which were not held constant across different levels of the other factors as follows: a factorial design: 1, 2, 4, 8, and 16 aphids for the first instar larva of ladybird; 2, 4, 8, 16, and 32 aphids for the second instar larva of ladybird; and 4, 8, 16, 32, and 64 aphids for the third and fourth instar larvae, and both female and male adults of ladybird, as the predation increases as the larva proceeds to the next stage [45]. There was a total of 120 treatments, and each treatment was repeated five times.

Each instar ladybird larva was starved for 12 h, and adults were starved for 24 h prior to experiment use. Predation was assessed by placing a single larva/adult in an experimental arena consisting of a petri dish (2 cm high, 10 cm diameter) containing 5 wheat leaves, approximately 8 cm long. After 24 h, the number of aphids that remained in each dish was recorded [30]. The bottom end of each wheat leaf was wrapped in a water-saturated absorbent cotton to provide sufficient moisture.

The predation amount (prey consumption, *N_e_*) = prey (aphid) densities (*N_0_*) − aphids remaining in the dish. The predation rate = (*N_e_/N_0_*). The relationship between ladybird predation and the initial number of aphids was analyzed by logistic regression to determine the functional response types of the ladybird. The relationship between *N_e_/N**_0_* and *N**_0_* fits with a polynomial function: *N_e_/N_0_* = exp (*P_0_* + *P_1_N_0_* + *P_2_N_0_^2^* + *P_3_N_0_^3^*)/(1 + exp (*P_0_* + *P_1_N_0_* + *P_2_N_0_^2^* + *P_3_N_0_^3^*) using the R software (R Core Team. 2021) and analysis for functional response modeling by the ‘frair’ package (version 0.5.100, date: 26 March 2017) [49,50], where *P_0_*, *P_1_*, *P_2_*, and *P_3_* are the intercept, linear, quadratic, and cubic coefficients, respectively. If *P_1_* > 0, significantly, and *P_2_* < 0, the functional response is type III, in which the proportion of prey consumed is positively density-dependent when prey densities are low. Meanwhile, if *P_1_* < 0, significantly, the functional response is type II, in which the proportion of prey consumed is inversely density-dependent [37,38,51,52].

If the logistic regression analysis indicated that the response function of *P. japonica* prey to *S. gramnium* is type II, then the random predator equation *N_e_* = *N_0_*(1 − exp(a*T_h_N*_e_ − a*T*)) [53] is used to further estimate the handling times (*T_h_*) and attack rate (*a*). Where *N*_e_ is the number of prey consumed, *N*_0_ is the initial prey density and *T* is the experimental duration (24 h). The functional response fits were nonparametrically bootstrapped (*n* = 2000) to obtain 95% confidence intervals (CIs) around the functional response curves and associated parameters. The random predator equation was then fitted to the bootstrapped dataset, with the initial parameter values estimated from the original ML estimates. If the confidence intervals between the two functional response curves merged, the functional responses and/or corresponding parameters were not significantly different. For each developmental stage of *P. japonica*, the functional response curves were compared for all pairwise combinations of wheat lines. 

To evaluate the biological control potential, the predatory efficiency (or net attack frequency) was estimated using *a*/*T_h_* [36]. 

### 4.5. Statistical Analysis

Raw data were collected using a WPS office software table and preprocessed. All parameters of the four wheat-acclimated aphid populations were calculated and analyzed using one-way ANOVA. The differences in *a*, *T_h_*, and *a*/*T_h_* among different age stages of ladybirds and their preying on wheat-acclimated aphid populations were analyzed using ANOVA based on a two-factor fixed-effects model. If the *p* < 0.05, the post hoc Tukey’s test was used to analyze the differences in the life-history trait for all treatments. All ANOVA were performed using SPSS 20.0 (SPSS Inc., Chicage, IL, USA).

## 5. Conclusions

Different wheat lines have diverse resistance traits to *S. graminium*. The wheat lines ‘98-10-19’ have a relatively higher resistance, and this resistance could further decrease the prey and proportion of the prey consumed by *P. japonica* to *S. graminum*, and these effects were different among the developmental stages and sexes of the predators, in which the male predator decreased most. However, this resistance did not negatively affect the biological control potential of *P. japonica* against *S. graminum*, especially in the fourth instar larvae of the predator.

## Figures and Tables

**Figure 1 plants-11-02754-f001:**
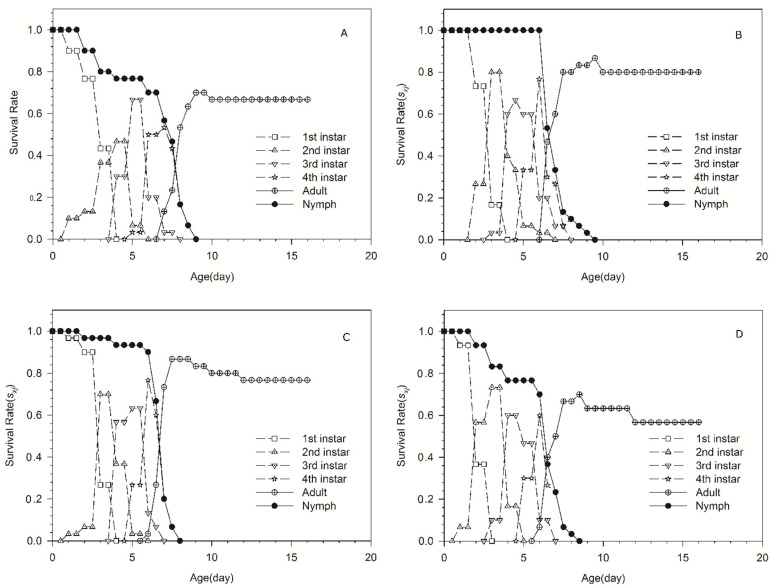
The age-stage survival rates of four *Schizaphis graminum* acclimated populations. (**A**) ‘98-10-19’-acclimated population; (**B**) ‘98-10-30’-acclimated population; (**C**) ‘186Tm39’-acclimated population; (**D**) ‘186Tm47’-acclimated population.

**Figure 2 plants-11-02754-f002:**
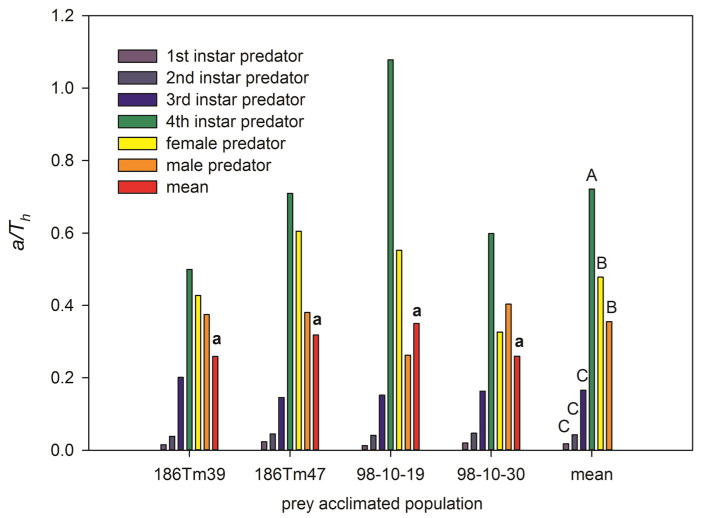
The predatory efficiency (or net attack frequency, *a/T_h_*) of *Propylaea japonica* compared among prey aphids from different acclimated *Schizaphis graminum* populations at different development stage of the predators. Lowercase letter ‘a’ above bars shows that the mean of *a/T_h_* has no significant differences among 4 acclimated populations at *p* < 0.05. Different capital letters above bars (rightmost group) are significant differences among the different age stages of enemy at *p* < 0.01.

**Table 1 plants-11-02754-t001:** The life-history traits of four wheat-acclimated *Schizaphis graminum* populations (mean ± SE).

Life-History Parameter	186Tm39	186Tm47	98-10-19	98-10-30	*F*	*df*	*p*
W1st (µg)	23.63 ± 0.80	21.93 ± 0.75	22.87 ± 0.77	21.30 ± 0.59	1.97	3, 116	0.123
Wa (µg)	324.4 ± 15.3 B	326.8 ± 12.0 B	264.2 ± 20.3 C	400.9 ± 16.8 A	9.09	3, 91	<0.001
DT (d)	6.12 ± 0.08 B	6.24 ± 0.15 B	7.05 ± 0.15 A	6.41 ± 0.14 B	10.80	3, 91	<0.001
MRGR	0.431 ± 0.012 A	0.437 ± 0.016 A	0.352 ± 0.020 B	0.459 ± 0.008 A	7.33	3, 91	<0.001
Fecundity	27.69 ± 1.95 BC	33.52 ± 2.66 AB	23.29 ± 2.59 C	38.48 ± 2.64 A	5.95	3, 91	0.001
r_m_	0.197 ± 0.007 A	0.204 ± 0.007 A	0.155 ± 0.012 B	0.202 ± 0.009 A	8.12	3, 91	<0.001
NRR	2.30 ± 0.17 AB	2.68 ± 0.19 A	1.71 ± 0.20 B	2.98 ± 0.19 A	11.79	3, 91	<0.001
NSR	0.867 ± 0.063	0.700 ± 0.085	0.700 ± 0.085	0.900 ± 0.056	2.11	3, 116	0.102

Note: W1st is the weight of 1st instar larvae born within 24 h, Wa is the weight of adult emergence within 24 h, DT is development time, MRGR is mean relative growth rate, r_m_ is intrinsic rates of natural increase, NRR is net reproduction rate, NSR is the nymph survival rate. The post hoc test was taken based on the results of ANOVA. Fecundity, r_m_ and NRR were significantly influenced by interaction between wheat line and generation. The different capital letters indicate the differences are significant among different wheat-acclimated aphid populations based on Tukey’s test (*p* < 0.01).

**Table 2 plants-11-02754-t002:** Three-way (wheat line, prey densities, and ladybird age stage) ANOVA results of the prey consumed (*N_e_**)* and proportion of prey consumed (*N_e_/N_0_*) of *P. japonica*.

		WL	PD	EAS	WL × PD	WL × EAS	PD × EAS	WL × PD × EAS
*N_e_*	*df*	3	5	6	15	18	18	54
	*F*	5.77	292.57	1816.27	3.13	2.48	81.18	1.46
	*p*	0.001	<0.001	<0.001	<0.001	0.001	<0.001	0.02
*N_e_/N* * _0_ *	*df*	3	5	6	15	18	18	54
	*F*	4.47	278.27	193.88	1.95	3.20	7.18	0.74
	*p*	0.004	<0.001	<0.001	0.02	<0.001	<0.001	0.91

Note: WL is wheat line (= wheat-acclimated aphid population), PD is prey density, EAS is enemy age stage.

## Data Availability

Not applicable.

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
