# Peer review of "Effects of Crop Resistance on the Tritrophic Interactions between Wheat Lines, Schizaphis graminum (Hemitera: Aphididae), and Propylaea japonica (Coleoptera: Coccinellidae)"

_plants, 2022, doi:10.3390/plants11202754_

Round 1

Reviewer 1 Report

Here, the authors address the tri-trophic interaction between different wheat lines, an aphid pest and their main predator. This is an interesting study object because potential synergies between the interaction partners can further improve pest control.

In general, the methods seem adequate to me, although sample size for each treatment was low (n = 5). Thus, I suggest to focus on the pooled data (across the different life-stages). The life-stage specific results and figures might be better placed in a supplementary material.

The introduction is not sufficient in content and the structure (intro, results, discussion, methods, conclusion) is a bit odd to me (see detailed comments below). The English language quality is rather poor (particularly in the method section) and requires thorough language editing!

Detailed comments:

L 13: “to control the pest sustainably” omit “the” and consider reformulating the sentence to “sustainably control pest…”

L 13: unclear to me what “persistent” means in this context – considering omitting this term

L 16-20: This results sentence is far too long and difficult to grasp – please split into different clear and shorter sentences.

L 20 onwards: please omit abbreviations in the abstract (Ne, a/Th, etc.)

Introduction: That is a very awkward way to start an introduction – your imagination is irrelevant in this context. In general, the introduction is ridiculously short and does not provide sufficient background to the topic, stat-of-the art for tri-trophic interactions with crop resistance, rational behind the hypothesis – please add more ‘meat to the bones’!

Table 2: Please state in the heading again what Ne and Ne/No are, so the reader can understand the table without having to check in the main text.

Tables 3 & 4/Figures 2-7: Please only show the data (table and figures) of the variables pooled across the different life-stages in the main text (No. of replicates for each treatment are low anyway). The tables and figures separated by stages should be added as supplementary material.

Discussion: The start of the discussion sounds like an introduction (i.e. could be moved to the introduction instead). Please start the discussion by shortly summarize your results and then elaborate on the specific findings and put them into a broader perspective/compare with former findings/relevant literature.

M & M: I found it a bit strange that the method sections is between Discussion and Conclusion. Is this according to the author guidelines for this journal?! Otherwise, I strongly suggest to put the methods between introduction and results.

I would have expected more background information about the resistance of the different crop lines. It should also be clear very early in the manuscript to what these lines are resistant.

Author Response

Comments and Suggestions for Authors

Reviewer 1:

Here, the authors address the tri-trophic interaction between different wheat lines, an aphid pest and their main predator. This is an interesting study object because potential synergies between the interaction partners can further improve pest control.

In general, the methods seem adequate to me, although sample size for each treatment was low (n = 5). Thus, I suggest to focus on the pooled data (across the different life-stages). The life-stage specific results and figures might be better placed in a supplementary material.

The introduction is not sufficient in content and the structure (intro, results, discussion, methods, conclusion) is a bit odd to me (see detailed comments below). The English language quality is rather poor (particularly in the method section) and requires thorough language editing!

Detailed comments:

L 13: “to control the pest sustainably” omit “the” and consider reformulating the sentence to “sustainably control pest…”

Re: Thanks, we have revised it (L13).

L 13: unclear to me what “persistent” means in this context – considering omitting this term

Re: Thanks, we have deleted it (L13).

L 16-20: This results sentence is far too long and difficult to grasp – please split into different clear and shorter sentences.

Re: Thanks, we have revised it(L 16-20).

L 20 onwards: please omit abbreviations in the abstract (Ne, a/Th, etc.)

Re: Thanks, we have deleted it(L 20,21,23).

Introduction: That is a very awkward way to start an introduction – your imagination is irrelevant in this context. In general, the introduction is ridiculously short and does not provide sufficient background to the topic, stat-of-the art for tri-trophic interactions with crop resistance, rational behind the hypothesis – please add more ‘meat to the bones’!

Re: Thanks, we have reworded the section of introduction.

Table 2: Please state in the heading again what Ne and Ne/No are, so the reader can understand the table without having to check in the main text.

Re: Thanks, we have revised it(L 67).

Tables 3 & 4/Figures 2-7: Please only show the data (table and figures) of the variables pooled across the different life-stages in the main text (No. of replicates for each treatment are low anyway). The tables and figures separated by stages should be added as supplementary material.

Re: Thanks, we have removed them from main text, and added as supplementary material.

Discussion: The start of the discussion sounds like an introduction (i.e. could be moved to the introduction instead).

Re: Thanks, we have moved that part to introduction.

Please start the discussion by shortly summarize your results and then elaborate on the specific findings and put them into a broader perspective/compare with former findings/relevant literature.

Re: Thanks, we have reworded these parts (lines233-240).

M & M: I found it a bit strange that the method sections is between Discussion and Conclusion. Is this according to the author guidelines for this journal?! (Yes, that we according to the author guidelines for this journals. ) Otherwise, I strongly suggest to put the methods between introduction and results.

Re: Thanks.

I would have expected more background information about the resistance of the different crop lines. It should also be clear very early in the manuscript to what these lines are resistant.

Re: Thanks, we have added more information in the introduction section(last paragraph of the introduction) (lines102-104).

In "Effects of crop resistance on the tritrophic interactions between wheat lines, Schizaphis graminum (Hemitera: Aphididae), and Propylaea japonica (Coleoptera: Coccinellidae)" the authors report the results of a study examining how acclimation of an important aphid pest to resistant and susceptible wheat germplasm affects the predation rate of an important biocontrol agent.

The manuscript is well presented, but there are some sections where the phrasing is a bit unclear. So I recommend that the authors also re-check some of their phrasing to ensure their points come across as clearly as possible.

Re: We are very grateful to the reviewer for her/he insightful suggestions and comments, we had carefully re-checked all of text. We hope our points come across as clearly as possible.

Reviewer 2 Report

In "Effects of crop resistance on the tritrophic interactions between wheat lines, Schizaphis graminum (Hemitera: Aphididae), and Propylaea japonica (Coleoptera: Coccinellidae)" the authors report the results of a study examining how acclimation of an important aphid pest to resistant and susceptible wheat germplasm affects the predation rate of an important biocontrol agent.

The manuscript is well presented, but there are some sections where the phrasing is a bit unclear. So I recommend that the authors also re-check some of their phrasing to ensure their points come across as clearly as possible.

General comments:

The authors use "wheat line" "wheat population" and "wheat variety" interchangeably, the authors should select one and use this consistently. Maybe "wheat line" is most appropriate. There is similar interchangability of terminology when discussing the aphid populations, I would suggest the authors use "aphid population" or "wheat-acclimiated aphid population".

Some of the figures and tables are unclear and difficult to interpret, I recommend that the authors display their results more clearly - I have provided some recommendations below. For example, using superscript for the results of the multiple comparisons would help with the interpretation of the tables.

Introduction:

Currently, the introduction is on the short side and could be expanded to better place the research in scientific context. For example, the auhtors could provide a more in depth introduction of the importance of wheat as a global crop, cover the economic importance of Schizaphis graminum as a pest species in more detail, discuss why it is important to develop aphid-resistant wheat lines and discuss why it is important to explore the tri-trophic wheat-aphid-predator relationship using contrasting wheat varieties.

In addition, the auhtors should use the introduction to make the case for testing acclimated aphid populations - why is it important that we investigate the effect aphid acclimiation to resistant/susceptible plants for ~12 months has on predator efficiency? This is the central question of the manuscript, but unfortunately a strong case for exploring this isn't made in the manuscript.

Results:

How do the authors know that these wheat lines represent a mixture of resistant and susceptible wheat lines? Has the resistnce status of these lines been described previously or is this the first description of these resistant lines? If this is the first description of the resistance status of these lines, then the authors should place more emphasis on this in the manuscript.

Currently the authors acclimate aphids to one of the plants over a year and assess the fitness of these acclimated aphids, a quick look at the resulting life-history data indicates that only one wheat line (98-10-19) is "resistant", with aphids shopwing higher development time, lower fecundity, and lower MRGR when feeding on this line; with all other lines, generally, showing little-to-no difference in aphid life-history and line 98-10-30 being the most susceptible. The authors should discuss this in more detail the manuscript.

Table 1: Some aspects of the table are currently unclear. It would aid readability if the authors renamed the "population" heading to "aphid population" or "wheat-acclimated aphid population" to avoid confusion with the different wheat lines, this title should also be placed this above the four columns that show the results for the four aphid lines, not to the side. The current location of the "population" heading should be replaced with "life-history parameter" as this is what is reported underneath. The units for the aphid weight measurements are missing.

Table 3: The title "aphid population" should be placed above the different populations

Figure 1: These graphs are unclear and difficult to interpret. Is this a survival curve, a development curve, or both? The y-axis (Survival) is a bit unclear, it suggests that all the 1st instar nymphs have died, when in fact most will have developed into adults. I suggest the authors chose to display either 1) survival data for the aphid over its life-time (i..e., a survival curve) or 2) the development time for the aphid to each development stage (i.e., a time to event curve). In either case, the graph could be made more reader friendly if all development stages were shown on the same line, with different points (circle, triangle, square etc.,.) indicating when each development stage is reached.

Table 4: Is "wheat line" the wheat line or the aphid-acclimiated aphid populaiton? Some sections are highlighted yellow.

Figures 2-7: Again, these are difficult to interpret. Am I correct in concluding that the authors are comparing across the different plant-acclimated aphid populations? If so they should attempt to display the results for each life-stage for each aphid population on the same panel, that way the reader can see the differences more easily. At the moment, the only message I can take away from the graphs is that the predators consume more aphids when there is a higher initial aphid density, it is difficult to see differences between the different plant-acclimated aphid populations. As these results are the key findings of the manuscript, I think it is important the authors maximise the clarity of these graphs.

Figure 8: This Figure appears at the end of the disucssion but isn't mentioned in the manuscript?

Materials and methods:

Some additional information on the experimental design is required.

Section 4.1: What growth stage of wheat was used for aphid rearing? 

Section 4.2: Specify the growth stage using a scientific scale (e.g., Feekes or Zadoks)

Section 4.3: The authors should explain how they caluclated the intrinsic rate of population increase (rm) and include the equation used to caluclate this parameter.

What experimental design was used? Factoral design or randomised design?

Section 4.4: By "peeling" do the authors mean "moulting"? This first sentence should also be restructured to improve clarity.

The authors need to describe how they calculated all of the predator-associated varibales (e.g., attack rate, prey consumption etc.,.). It is clear how the proportion of prey consumed was caluclated, but the method used to calculate the predator attack rate is not described. 

Section 4.5:  What is "WPS"?

I would argue that the methods and equations used to caluclate the various parameters measured belong in the appropriate methods sections that describe the associated experiment, not in the statistics section which should just detail the statistical methodology used.

Should development time/aphid sruvival not have been analysed using a survival or time-until-event analysis?

Conclusion:

I don't think the authors can conclude that their results show "sustained" resistance is present in the wheat lines tested as the experiment only acclimated the aphids for one year, which is not sufficient time for the aphids to evolve mechanisms to break this resistance.

Author Response

Reviewer 2

General comments:

The authors use "wheat line" "wheat population" and "wheat variety" interchangeably, the authors should select one and use this consistently. Maybe "wheat line" is most appropriate. There is similar interchangability of terminology when discussing the aphid populations, I would suggest the authors use "aphid population" or "wheat-acclimiated aphid population".

Re: Thanks, we have revised it.

Some of the figures and tables are unclear and difficult to interpret, I recommend that the authors display their results more clearly - I have provided some recommendations below. For example, using superscript for the results of the multiple comparisons would help with the interpretation of the tables.

Re: Thanks, we have reword them.

Introduction:

Currently, the introduction is on the short side and could be expanded to better place the research in scientific context. For example, the auhtors could provide a more in depth introduction of the importance of wheat as a global crop, cover the economic importance of Schizaphis graminum as a pest species in more detail, discuss why it is important to develop aphid-resistant wheat lines and discuss why it is important to explore the tri-trophic wheat-aphid-predator relationship using contrasting wheat varieties.

Re: Thanks, we have reworded this section.

In addition, the auhtors should use the introduction to make the case for testing acclimated aphid populations - why is it important that we investigate the effect aphid acclimiation to resistant/susceptible plants for ~12 months has on predator efficiency? This is the central question of the manuscript, but unfortunately a strong case for exploring this isn't made in the manuscript.

Re: Thanks, we have revised it (Lines 92-97).

Results:

How do the authors know that these wheat lines represent a mixture of resistant and susceptible wheat lines? Has the resistance status of these lines been described previously or is this the first description of these resistant lines? If this is the first description of the resistance status of these lines, then the authors should place more emphasis on this in the manuscript.

Re: Thanks, our previous work show these four wheat line have difference to Sitobion avenae, but we do not know the resistance status to greenbug. We have added this information in the manuscript (Lines 102-105).

Currently the authors acclimate aphids to one of the plants over a year and assess the fitness of these acclimated aphids, a quick look at the resulting life-history data indicates that only one wheat line (98-10-19) is "resistant", with aphids showing higher development time, lower fecundity, and lower MRGR when feeding on this line; with all other lines, generally, showing little-to-no difference in aphid life-history and line 98-10-30 being the most susceptible. The authors should discuss this in more detail the manuscript.

Re: Thanks, we discuss this in the Lines 235-252.

Table 1: Some aspects of the table are currently unclear. It would aid readability if the authors renamed the "population" heading to "aphid population" or "wheat-acclimated aphid population" to avoid confusion with the different wheat lines, this title should also be placed this above the four columns that show the results for the four aphid lines, not to the side. The current location of the "population" heading should be replaced with "life-history parameter" as this is what is reported underneath. The units for the aphid weight measurements are missing.

Re: Thanks, we have revised the table.

Table 3: The title "aphid population" should be placed above the different populations

Re: Thanks, we have revised the table.

Figure 1: These graphs are unclear and difficult to interpret. Is this a survival curve, a development curve, or both? The y-axis (Survival) is a bit unclear, it suggests that all the 1st instar nymphs have died, when in fact most will have developed into adults. I suggest the authors chose to display either 1) survival data for the aphid over its life-time (i..e., a survival curve) or 2) the development time for the aphid to each development stage (i.e., a time to event curve). In either case, the graph could be made more reader friendly if all development stages were shown on the same line, with different points (circle, triangle, square etc.,.) indicating when each development stage is reached.

Re: Thanks, we have revised it.

Fig. 1. gives the probability (The y-axis) that a newborn will survive to age x and stage j. So, it did not suggests that all the 1st instar nymphs have died, but the survival rate: some of them died, but some of them molted and become 2nd instar nymphs, and so on.... There are significant overlaps between curves of different stages. For any age x, a newborn can survive only to one of the stages, therefore, it is always true that in Fig. 1.

Some researchers have ignored the variable developmental rate and have used the rounded means of each stage to divide the life span into nonoverlapping stages. Chi (1988) discussed the problems and errors in ignoring stage overlapping. So we retained fig.1, and show time and survival data of each development stage the in supplementary material.

Table 4: Is "wheat line" the wheat line or the aphid-acclimiated aphid populaiton? Some sections are highlighted yellow.

Re: Thanks, we have revised the table.

Figures 2-7: Again, these are difficult to interpret. Am I correct in concluding that the authors are comparing across the different plant-acclimated aphid populations? If so they should attempt to display the results for each life-stage for each aphid population on the same panel, that way the reader can see the differences more easily. At the moment, the only message I can take away from the graphs is that the predators consume more aphids when there is a higher initial aphid density, it is difficult to see differences between the different plant-acclimated aphid populations. As these results are the key findings of the manuscript, I think it is important the authors maximise the clarity of these graphs.

Re: Thanks, , we have revised the section.

Figure 8: This Figure appears at the end of the disucssion but isn't mentioned in the manuscript?

Re: Thanks, we have mentioned it in result section (line 224, and 289).

Materials and methods:

Some additional information on the experimental design is required.

Section 4.1: What growth stage of wheat was used for aphid rearing? 

Re: Thanks, we have added these information (lines313-314).

Section 4.2: Specify the growth stage using a scientific scale (e.g., Feekes or Zadoks)

Re: Thanks, we have added these information (line317).

Section 4.3: The authors should explain how they caluclated the intrinsic rate of population increase (rm) and include the equation used to caluclate this parameter.

Re: Thanks, we have adjusted these information from Section “4.5. Statistical analysis” To Section “4.3:”

What experimental design was used? Factoral design or randomised design?

Re: Thanks, we have added these information (lines 321-331).

Section 4.4: By "peeling" do the authors mean "moulting"? This first sentence should also be restructured to improve clarity.

Re: Thanks, we have revised it (line 351)

The authors need to describe how they calculated all of the predator-associated varibales (e.g., attack rate, prey consumption etc.,.). It is clear how the proportion of prey consumed was caluclated, but the method used to calculate the predator attack rate is not described. 

Re: Thanks, we have adjusted these information from “4.5. Statistical analysis” To Section “4.4:” and describe how to calculate “attack rate” in line 385, and “prey consumption” in line 371.

Section 4.5:  What is "WPS"?

Re: Thanks, we have revised it as “WPS office table”

I would argue that the methods and equations used to caluclate the various parameters measured belong in the appropriate methods sections that describe the associated experiment, not in the statistics section which should just detail the statistical methodology used.

Re: Thanks, we have adjusted those sections.

Should development time/aphid sruvival not have been analysed using a survival or time-until-event analysis?

Re: The results show in Fig. 1. And we added the results analysis in the text.

Conclusion:

I don't think the authors can conclude that their results show "sustained" resistance is present in the wheat lines tested as the experiment only acclimated the aphids for one year, which is not sufficient time for the aphids to evolve mechanisms to break this resistance.

Re: Thanks, we have revised these section.

Round 2

Reviewer 1 Report

Thanks for implementing many of my suggestions (extending the introduction, removing tables and figures to the supplementary material, improving the discussion ect.) in the revised version of the manuscript. Still, there are many language issues (see some initial comments below; However, it is beyond my service to edit the language…). The manuscript would greatly benefit from language editing by a native speaker.

Detailed comments:

L. 19: “for more than…”

L. 20: unclear to what “that” refers to – provide names whenever possible

L. 35-36: see my previous comment; please change to “are complicated.” and remove the “than we have imagined”

L. 36-37: change “consider” to “considered”

L. 40: change “be using” to “are used”

L. 45: “be consider” to “is considered”

L. 46-47: hard to understand – please reformulate

L. 51-53: Not really a “theoretical basis” but more expectations/or empirical evidence; Similarly: “been proved” is to hard (mathematical/logical) – better it has been shown/provided evidence for… (there are for sure also negative examples…) (The same in L. 61).

Please carefully check the rest of the MS or better let a native speaker edit the language!

Author Response

Respone to Review 1:

Open Review

(x) I would not like to sign my review report

( ) I would like to sign my review report

English language and style

( ) Extensive editing of English language and style required

(x) Moderate English changes required

( ) English language and style are fine/minor spell check required

( ) I don't feel qualified to judge about the English language and style

Yes Can be improved Must be improved Not applicable

Does the introduction provide sufficient background and include all relevant references?

( ) (x) ( ) ( )

Are all the cited references relevant to the research?

( ) (x) ( ) ( )

Is the research design appropriate?

( ) (x) ( ) ( )

Are the methods adequately described?

(x) ( ) ( ) ( )

Are the results clearly presented?

(x) ( ) ( ) ( )

Are the conclusions supported by the results?

( ) (x) ( ) ( )

Comments and Suggestions for Authors

Thanks for implementing many of my suggestions (extending the introduction, removing tables and figures to the supplementary material, improving the discussion ect.) in the revised version of the manuscript. Still, there are many language issues (see some initial comments below; However, it is beyond my service to edit the language…). The manuscript would greatly benefit from language editing by a native speaker.

Detailed comments:

  1. 19: “for more than…”
  2. 20: unclear to what “that” refers to – provide names whenever possible

Re: Thank you very much. We have revised it as “Results show that aphid fitness was lowest for aphids that had been feeding on wheat line '98-10-19' for one year. These aphids had the longest development time, lowest adult mass, lowest mean relative growth rate, and the lowest measurements of reproductive fitness. In contrast, aphids feeding on line '98-10-30' were the fittest aphids, with the shortest development time and highest levels of reproductive fitness”.

  1. 35-36: see my previous comment; please change to “are complicated.” and remove the “than we have imagined”

Re: Thank you. We have revised it.

  1. 36-37: change “consider” to “considered”

Re: Thank you. We have revised it.

  1. 40: change “be using” to “are used”

Re: Thank you. We have revised it.

  1. 45: “be consider” to “is considered”

Re: Thank you. We have revised it.

  1. 46-47: hard to understand – please reformulate

Re: Thank you very much. We have revised it as “The interactions between crop resistance and biological control is an area of growing research interest in an integrated pest management (IPM) strategy”.

  1. 51-53: Not really a “theoretical basis” but more expectations/or empirical evidence; Similarly: “been proved” is to hard (mathematical/logical) – better it has been shown/provided evidence for… (there are for sure also negative examples…) (The same in L. 61).

Re: Thank you very much. We have revised it.

Please carefully check the rest of the MS or better let a native speaker edit the language!

Re: Thank you very much. Our MS have been edited by Editage (www.editage.cn) for English language improve.

Submission Date

15 August 2022

Date of this review

21 Sep 2022 17:15:04

Reviewer 2 Report

The authors have made satisfactory improvements to the manuscript, taking most of the comments raised by myself and the other reviewer into account.

In addition, there are still some issues with the quality of the text and the phrasing of some sentences that make parts of the manuscript difficult to follow. Here the manuscript would benefit from additional improvements to clarity. Some examples from the abstract:

- Line 16-19: Rephrasing to "Results show that aphid fitness was lowest for aphids that had been feeding on wheat line '98-10-19' for one year. These aphids had the longest development time, lowest adult mass, lowest mean relative growth rate, and the lowest measurements of reproductive fitness. In contrast, aphids feeding on line '98-10-30' were the fittest aphids, with the shortest development time and highest levels of reproductive fitness".

- Line 19: Should be "mean relative growth rate" with "ration" removed. This needs to be changed throughout the manuscript.

- Line 20+ (description of the predator results): The messaging in this sentence is very unclear. 

Other points:

Throughout the manuscript, "wheat-acclimated aphid population" is sometimes misspelled as "wheat-acclaimed aphid population"

Line 71-72: The authors state "the grain aphid Sitobion miscanthi Takahashi, (= S. avenae (Fab.) in China)". This is incorrect. S. avenae and S. mischanti are two distinctive species (https://www.cabi.org/isc/datasheet/51740), unless the authors are stating that S. miscanthi replaces S. avenae as a top aphid pest in China. If this is the case, this would again be another example of why the authors need to greatly improve the quality of their text, to avoid confusing the reader like this.

In conclusion, the authors need to greatly improve the quality of the messaging in the text. In the current state, it is very difficult to even identify the main take-home messages from the manuscript or even interpret the meaning behind individual sentences.

Author Response

Response to Review 2:

Open Review

(x) I would not like to sign my review report

( ) I would like to sign my review report

English language and style

(x) Extensive editing of English language and style required

( ) Moderate English changes required

( ) English language and style are fine/minor spell check required

( ) I don't feel qualified to judge about the English language and style

Yes Can be improved Must be improved Not applicable

Does the introduction provide sufficient background and include all relevant references?

( ) (x) ( ) ( )

Are all the cited references relevant to the research?

(x) ( ) ( ) ( )

Is the research design appropriate?

(x) ( ) ( ) ( )

Are the methods adequately described?

( ) (x) ( ) ( )

Are the results clearly presented?

( ) (x) ( ) ( )

Are the conclusions supported by the results?

( ) (x) ( ) ( )

Comments and Suggestions for Authors

The authors have made satisfactory improvements to the manuscript, taking most of the comments raised by myself and the other reviewer into account.

Re: Thank you

In addition, there are still some issues with the quality of the text and the phrasing of some sentences that make parts of the manuscript difficult to follow. Here the manuscript would benefit from additional improvements to clarity. Some examples from the abstract:

- Line 16-19: Rephrasing to "Results show that aphid fitness was lowest for aphids that had been feeding on wheat line '98-10-19' for one year. These aphids had the longest development time, lowest adult mass, lowest mean relative growth rate, and the lowest measurements of reproductive fitness. In contrast, aphids feeding on line '98-10-30' were the fittest aphids, with the shortest development time and highest levels of reproductive fitness".

Re: Thank you very much. We have revised it.

- Line 19: Should be "mean relative growth rate" with "ration" removed. This needs to be changed throughout the manuscript.

Re: Thank you very much. We have revised it.

- Line 20+ (description of the predator results): The messaging in this sentence is very unclear.

Re: Thank you very much. We have revised it as “The prey consumed and proportion of prey consumed of predatory, especially male adult, were smallest significantly when they feeding the aphids of '98-10-19' acclimated population.”

Other points:

Throughout the manuscript, "wheat-acclimated aphid population" is sometimes misspelled as "wheat-acclaimed aphid population"

Re: Thank you so much. I feel horrible for my mistake. We have revised it.

Line 71-72: The authors state "the grain aphid Sitobion miscanthi Takahashi, (= S. avenae (Fab.) in China)". This is incorrect. S. avenae and S. mischanti are two distinctive species (https://www.cabi.org/isc/datasheet/51740), unless the authors are stating that S. miscanthi replaces S. avenae as a top aphid pest in China. If this is the case, this would again be another example of why the authors need to greatly improve the quality of their text, to avoid confusing the reader like this.

Re: Thank you so much. We have revised it as “Sitobion miscanthi Takahashi, (widely mis-reported as S. avenae (Fab.) in China),”

In conclusion, the authors need to greatly improve the quality of the messaging in the text. In the current state, it is very difficult to even identify the main take-home messages from the manuscript or even interpret the meaning behind individual sentences.

Re: Thank you so much. Our MS have been edited by Editage (www.editage.cn) for English language and the quality of the messaging improve in the text .

Submission Date

15 August 2022

Date of this review

26 Sep 2022 15:57:17
